# Genetic Risk Factors and Gene–Lifestyle Interactions in Gestational Diabetes

**DOI:** 10.3390/nu14224799

**Published:** 2022-11-13

**Authors:** Tiina Jääskeläinen, Miira M. Klemetti

**Affiliations:** 1Department of Food and Nutrition, University of Helsinki, P.O. Box 66, 00014 Helsinki, Finland; 2Department of Medical and Clinical Genetics, University of Helsinki, P.O. Box 63, 00014 Helsinki, Finland; 3Department of Obstetrics and Gynecology, Helsinki University Hospital, University of Helsinki, P.O. Box 140, 00029 Helsinki, Finland

**Keywords:** gestational diabetes, genetic background, polygenic risk score, gene–environment interaction

## Abstract

Paralleling the increasing trends of maternal obesity, gestational diabetes (GDM) has become a global health challenge with significant public health repercussions. In addition to short-term adverse outcomes, such as hypertensive pregnancy disorders and fetal macrosomia, in the long term, GDM results in excess cardiometabolic morbidity in both the mother and child. Recent data suggest that women with GDM are characterized by notable phenotypic and genotypic heterogeneity and that frequencies of adverse obstetric and perinatal outcomes are different between physiologic GDM subtypes. However, as of yet, GDM treatment protocols do not differentiate between these subtypes. Mapping the genetic architecture of GDM, as well as accurate phenotypic and genotypic definitions of GDM, could potentially help in the individualization of GDM treatment and assessment of long-term prognoses. In this narrative review, we outline recent studies exploring genetic risk factors of GDM and later type 2 diabetes (T2D) in women with prior GDM. Further, we discuss the current evidence on gene–lifestyle interactions in the development of these diseases. In addition, we point out specific research gaps that still need to be addressed to better understand the complex genetic and metabolic crosstalk within the mother–placenta–fetus triad that contributes to hyperglycemia in pregnancy.

## 1. Gestational Diabetes Impacts Global Health across Generations

Significant maternal metabolic adaptations are required during normal pregnancy to cater to the nutritional needs of the growing fetus. These include, for example, a progressive increase in insulin resistance in the second and third trimesters and upregulation of pancreatic insulin secretion [1]. When these physiologic changes or requirements occur on a background of chronic maternal insulin resistance and/or defects in maternal beta cell function, maternal hyperglycemia may develop. Gestational diabetes (GDM)—defined as maternal hyperglycemia with onset or first recognition during pregnancy excluding overt diabetes—affects a growing number of pregnancies worldwide and is the most common endocrinological complication in pregnant women [2]. However, reflecting the multifaceted role of environmental and genetic factors in GDM pathogenesis, as well as the prevailing lack of global consensus on diagnostic criteria, internationally reported rates vary widely [3,4]. A recent systematic review and meta-analysis reported a GDM prevalence of ≈15% using the IADPSG criteria [5].

Increasing frequencies of GDM and maternal obesity have significant public health repercussions. Both are independent and cumulative risk factors of adverse obstetric and perinatal outcomes, including cesarean section deliveries, fetal macrosomia, and maternal hypertensive disorders [6,7,8]. In addition, GDM is associated with later morbidity in both the mother and child [9,10,11,12]. A systematic review and meta-analysis reported that women with a history of GDM have nearly 10 times higher risk of developing type 2 diabetes (T2D) postpartum than women with normoglycemic pregnancies [13]. In a recent Finnish study with a 23-year follow-up, the prevalence of T2D increased linearly over time, reaching 50.4% in women with prior GDM (vs. 5.5% in women without GDM) by the end of the study period [14]. Type 1 diabetes (T1D) developed in 5.7% of Finnish women within 7 years after a GDM pregnancy, whereas no T1D cases were identified in the control group without GDM [14]. GDM also indicates an increased predisposition to cardiovascular diseases [15]. In fact, maternal glucose levels display a continuous relationship with later adverse cardiovascular risk factor phenotypes, even below GDM diagnostic thresholds [16]. In utero exposure to GDM, on the other hand, is associated with an increased risk of obesity, metabolic syndrome, and diabetes in the offspring [11,12,17,18]. Recently, the HAPO Follow-up Study (10–14 years after pregnancy) showed that, across the maternal glucose spectrum including the normoglycemia range, higher glucose levels are associated with higher childhood glucose and insulin resistance, independent of maternal and childhood BMI and family history of diabetes [19]. A similar relationship has been shown between maternal glycemia and childhood adiposity [20]. 

## 2. Heterogeneity of Gestational Diabetes Poses Clinical Challenges

In the past decades, a growing body of evidence has emerged on both phenotypic and genotypic heterogeneity in women with GDM [21,22,23]. In addition to obese and lean phenotypes [22], significant variation exists between individuals with GDM with respect to glycemic physiology and adaptation to pregnancy-associated changes in metabolism. The primary defect leading to pregnancy hyperglycemia may be excess insulin resistance, inadequate insulin secretion, or both [24], and it seems that obstetric and perinatal outcomes vary to some extent between physiologic GDM subtypes. For instance, higher frequencies of complications such as fetal macrosomia and cesarean sections have been reported in women with “insulin-resistant GDM” [25,26,27,28]. Variation also exists in terms of the time of onset of maternal hyperglycemia. Conventionally, GDM screening is performed in late pregnancy, between 24 and 30 weeks’ gestation, when gestation-related insulin resistance has set in. However, signs of abnormal metabolism can be identified in GDM women already before pregnancy or in early pregnancy [29,30,31]. Moreover, increasing data suggest that derangements in the maternal metabolic milieu—such as hyperinsulinemia—may affect placental and fetal development already from conception [32,33,34,35]. “Early GDM” (diagnosed <20 weeks’ gestation) may associate with even poorer pregnancy outcomes than the late-onset form of the disease [36,37]. However, there is a gap in research evidence concerning normal and abnormal glucose metabolism in early pregnancy [37,38], with no evidence-based criteria for the diagnosis of early-onset GDM. Finally, a possible source of heterogeneity is maternal ethnicity, which also appears to result in not only phenotypic but also genotypic differences between women with GDM [39].

Heterogeneity of GDM in terms of its time of onset, severity, and pathogenesis poses challenges with respect to its clinically meaningful screening and treatment, as complication frequencies and responsiveness to interventions (e.g., lifestyle changes) may vary between individuals [24,40,41]. Current treatment protocols do not differentiate between GDM subtypes. However, mapping the genetic architecture of GDM and accurate phenotypic and genotypic definition of GDM could potentially help in the individualization of GDM treatment and assessment of long-term prognoses [2,21].

## 3. Maternal Genetic Risk Factors of GDM

In the 1980s and 1990s, studies revealed familial clustering of both T1D and T2D as well as impaired glucose tolerance in families with GDM [42,43,44]. Subsequently, candidate gene studies showed that specific genetic variants associated with diabetes in non-gravid populations are more prevalent in women who develop GDM. In particular, variants of T2D-associated genes, such as *TCF7L2*, *KCNJ11*, *KCNQ1*, *CDKAL1*, *CDKN2A-CDKNA2B*, *FTO*, *HHEX*, *IGF2BP2*, *SLC30A8*, *MTNR1B*, and *PPARG*, have been observed more frequently in women with GDM, as compared to women with normoglycemic pregnancies, in various populations [45,46,47,48,49,50,51,52,53]. In addition, certain HLA alleles associated with T1D [54,55,56] and common polymorphism of the maturity-onset diabetes of the young (MODY) genes, such as GCK (MODY2 gene) and *HNF1A* (MODY3 gene), have been reported to associate with GDM [57,58]. 

More recently, the results of meta-analyses and genome-wide association studies (GWAS) have confirmed the association of certain T2D genetic variants with GDM. A list of genes associated with GDM in recent meta-analyses, systematic reviews, or GWAS are shown in Table 1. To date, three GWAS have been performed on genetic variants predisposing to GDM. The first GWAS, by Kwak et al. on Korean women, identified the association of *CDKAL1* and *MTNR1B* with GDM [59]. Another GWAS with a more limited sample size was not able to identify loci at a genome-wide significant level [60]. The largest GWAS to date was recently performed by Pervjakova et al. [61], involving 5485 women with GDM and 347,856 controls without GDM. Utilizing a multi-ancestry meta-analysis, this study confirmed the associations of T2D-linked genes *MTNR1B*, *TCF7L2*, *CDKAL1*, and *CDKN2A-CDKN2B* with GDM. In addition, *HKDC1* emerged as a locus that is more strongly associated with GDM than with T2D in non-pregnant populations [61]. Interestingly, most genetic variants showing robust association with GDM are variants with confirmed or hypothesized association with β-cell function or insulin secretion [62]. Of the T2D genetic loci identified to date, *MTNR1B* has shown the strongest reported association with GDM [63], whereas *TCF7L2* is the locus most strongly associated with T2D [64]. With regards to insulin-resistance-associated T2D-loci, only *IRS* variants have demonstrated significant association with GDM in GWAS or meta-analyses (Table 1), whereas *PPARG* has been reported to associate with GDM only in candidate gene studies [52]. Although obesity is a common risk factor for GDM, fat-mass/obesity-associated gene variants that predispose to T2D (e.g., *FTO*) have been linked to GDM in candidate gene studies [52,65,66] but not yet in GWAS.

During the past ten years, studies have also shed light on some of the genetic factors that regulate maternal glycemic traits during the metabolically exceptional period of pregnancy. In a study involving 5517 women of European and South Asian origin, Freathy et al. [50] showed that a maternal *GCK* variant is associated with higher fasting glucose in European and Thai women, 1h post-load glucose in European women, and 2 h post-load glucose in Thai women in an OGTT at ≈28 weeks’ gestation. In the same study, a *TCF7L2* variant was shown to associate with both fasting- and post-load glucose concentrations during a late-pregnancy OGTT in Europeans but not in Thais [50]. Hayes et al. [67], on the other hand, explored genetic factors that influence maternal glycemic traits in a GWAS setting. This study identified four genetic variants (*G6PC2* (association with fasting plasma glucose); *MTNR1B* (association with fasting and 1 h post-load plasma glucose); *GCKR* (association with fasting plasma glucose and C-peptide); and *PP1R3B* (association with fasting plasma glucose)) that have been previously associated with glycemic traits or T2D in non-pregnant populations, in addition to two novel variants, *HKCD1* (hexokinase-domain-containing 1, associated with 2 h post-load plasma glucose) and *BACE2* (β-site amyloid polypeptide-cleaving enzyme 2, associated with fasting C-peptide), which have shown only weak associations with T2D [67]. The *HKCD1* variant has been associated with post-load glucose concentrations in non-pregnant populations [68], and the *BACE2* variant has been implicated in β-cell mass and function [69]. Considering that the *HKCD1* locus was also reported to be associated with GDM in the recent GWAS by Pervjakova et al. [61], it is likely to be relevant concerning the regulation of gestational glycemia [67]. Variants linked to maternal hyperglycemia appear to reduce the expression of *HKDC1* [70], a hexokinase with a suggested role in whole-body glucose homeostasis [71]. 

Although multiple common genetic variants have so far been linked with GDM, it should be taken into account that the effect sizes of individual variants on GDM risk are relatively low, underlining the significance of environmental factors in GDM. We cannot exclude the possibility that, in the future, the discovery of low-frequency or rare variants with large effects may provide further insight into the genetic etiology of GDM [72]. However, even larger consortium studies should be carried out in the future to reach the statistical power needed to identify rarer genetic factors that may predispose to or protect against GDM. 

## 4. Fetal, Paternal and Placental Genetic Risk Factors of GDM

Although current antenatal evaluation of GDM risk in pregnant women only involves the assessment of maternal characteristics, it is likely that also fetal, paternal, and/or placental factors affect its development and contribute to the observed heterogeneity in terms of obstetric and fetal outcomes [21]. The role of fetoplacental factors is suggested by epidemiological studies showing that GDM risk and/or maternal glycemia are affected by, e.g., fetal sex [73], multiple pregnancies [74], paternal age [75], and paternal ethnicity [76]. Moreover, a recent meta-analysis reported that GDM reoccurs only in approximately half of subsequent pregnancies [77]—less than what could be expected if GDM were only a sign of deranged glucose metabolism in the mother [21]. Nevertheless, relatively few studies to date have investigated paternal, fetal, and placental genetic factors that affect maternal glucose metabolism and/or GDM risk. Petry et al. have reported that certain paternally transmitted fetal *IGF2* polymorphisms are associated with elevated maternal glucose concentrations in pregnancy, possibly via changes in placental IGF2 expression [78]. Hivert et al., on the other hand, showed that placental DNA methylation affects the regulation of maternal insulin sensitivity in pregnancy [79]. One potential mechanism by which placental metabolism and gene expression could have an impact on glucose metabolism in maternal tissues is via extracellular vesicles (EV), a newly identified means of cell–cell communication [80,81,82]. Recent studies have already identified some differences in the size, concentration, and cargo of circulating placental EVs in women with GDM, as compared to women with normoglycemic pregnancies [82], and suggested potential biological effects in maternal target tissues [83]. However, the mechanisms of molecular and genetic crosstalk within the mother–placental–fetus triad, implicated in glucose homeostasis during pregnancy, are a largely unknown field that should be addressed in future research. Filling this gap in knowledge requires the collection of placental, cord blood, and paternal samples in large diverse cohorts.

## 5. Polygenetic Risk Scores (PRSs)

As highlighted earlier, GDM has a highly polygenic architecture. A set of risk variants can be combined into a polygenic risk score (PRS) providing an estimated effect of genetic risk information from across the genome [105]. PRS typically represents a weighted sum of the number of risk alleles carried by an individual, wherein the risk alleles and their weights are defined by the loci and their measured effects as detected by GWAS.

Since GDM and T2D share similar pathophysiology and genetic susceptibility, the PRSs investigated thus far have mostly been constructed by using known variants associated with T2D (Table 2). Most of the SNPs used in these PRS studies have typically been robustly associated (*p* < 5 × 10^−8^) with T2D in many GWASs in the general population. 

### 5.1. PRSs and GDM

In many of the PRS studies performed thus far, women with GDM appear to have higher PRSs than controls, but the discriminative ability of PRSs for GDM has been modest. However, it appears that PRSs may improve GDM risk stratification for some women, especially when combined with clinical parameters. Interestingly, a recent retrospective study of 1270 women with self-reported GDM and 13,400 controls utilized a genetics-based screening tool to identify women at risk for GDM, even before they became pregnant [106]. A PRS based on 84 genetic variants, analyzed from saliva or a cheek swab test, predicted that women in the top 5% of PRS have more than a sixfold increased risk of GDM compared to the lower 50% of the PRS.

### 5.2. PRSs and T2D

PRSs have also been shown to be predictive of pre-diabetes and T2D among women with prior GDM years, after delivery (Table 2). Some studies have observed that a PRS in conjunction with conventional risk factors for T2D results in improved discrimination of the risk of pre-diabetes in women with prior GDM, likely providing more accurate tools for the prediction of future T2D [107]. However, it should be taken into account that some of these studies are limited by small numbers of T2D cases and relatively short follow-up periods. Moreover, we should acknowledge the fact that many of the genetic variants included in the PRSs are not confirmed to be associated with T2D in all populations.

**Table 2 nutrients-14-04799-t002:** Polygenic risk scores studied in women with gestational diabetes.

Reference	Population	Polygenic Risk Score (PRS)	Outcome: GDM	Results and Effect (If Available)
Kawai et al. [108]	458 women with GDM and 1538 controls	34 SNPs previously associated with T2D or fasting glucose in the general population, or with GDM or glucose intolerance in pregnancy	GDM	PRS associated with GDM. 1.11 (1.08–1.14) per risk allele increase.
Lauenborg et al. [52]	244 women with GDM and 1883 controls	11 SNPs previously associated with T2D	GDM	PRS associated with GDM. 1.18 (1.10–1.27) per risk allele increase.
Powe et al. [109]	250 women with GDM and 1681 controls (2 separate cohorts: HAPO and Gen3G)	150 SNPs previously associated with glycemic traits or T2D	GDM	Both PRSs associated with GDM. 1.06 (1.01–1.10) for Gen3G and 1.03 (1.01–1.06) for HAPO per risk allele.
			Outcome: T2D	
Ekelund et al. [110]	793 women with GDM	13 SNPs previously associated with T2D	T2D	PRS associated with T2D. 1.11 (1.05–1.18) per risk allele increase.
Kwak et al. [111]	395 women with GDM	48 SNPs previously associated with T2D	T2D	PRS associated with T2D. 1.66 (1.30–2.13) per risk allele increase.
Li at al. [112]	2434 women with GDM	59 SNPs previously associated with T2D	T2D	PRS associated with T2D. 1.07 (1.01–1.14) per 5 risk alleles.
			Outcome: GDM and T2D	
Cormier et al. [107]	214 women with GDM and 82 controls	36 SNPs previously associated with T2D	GDM, pre-diabetes, and T2D	PRS associated with GDM and progression to pre-diabetes and T2D.
Sullivan et al. [113]	281 women with GDM and 1102 controls	34 SNPs previously associated with T2D	GDM and T2D	PRS associated with previous GDM, but not with T2D. 1.05 (1.00–1.08) per risk allele increase for GDM.
			Outcome: other glycemic traits	
Prasad et al. [114]	374 women with GDM	4 SNPs associated with measures of insulin resistance and secretion	Insulin secretion, insulin resistance, and T2D	PRS associated with impaired insulin secretion (disposition index) and resistance (HOMA-IR)

GDM = gestational diabetes; SNP = single-nucleotide polymorphism; T2D = type 2 diabetes.

Recently, studies have also examined the associations of PRSs with maternal glycemic traits. Powe et al. showed utilizing data from two cohorts (Genetics of Glucose Regulation in Gestation, *n* = 573, and Hyperglycemia and Adverse Pregnancy Outcomes (HAPO), *n* = 4431) that genetic risk scores for elevated fasting glucose and insulin, reduced insulin secretion and sensitivity, and T2D, built using SNPs linked to glycemic traits in non-pregnant populations, were also associated with these traits during pregnancy [109]. In a more recent novel study, involving more than 5000 women, Powe et al. utilized Bayesian nonnegative matrix factorization (bNMF) clustering techniques to identify genetic variants associated with physiologic pathways implicated in GDM [115]. They selected 222 SNPs associated with T2D [116] and 4 SNPs (from loci near *HKDC1*, *G6PC2*, *PCSK1*, *PPP1R3B*) associated with glycemic traits during pregnancy [67] at a genome-wide level. The clusters that emerged were labeled on the basis of their physiologic linkages as “β-cell”, “obesity”, “proinsulin”, “lipodystrophy”, and “liver-lipid” cluster polygenic scores, in addition to five pregnancy-related clusters. Associations were found between GDM and cluster polygenic scores involving genetic determinants of impaired β-cell function and hepatic lipid metabolism [115]. Moreover, a pregnancy-physiology-based cluster (including higher post-load glucose, lower disposition index, and higher adiposity measures as the strongest weighted maternal characteristics) was associated with GDM and higher birth weight [115]. In contrast, the “β-cell cluster” including variants linked to decreased insulin secretion was associated with reduced birth weight, in concordance with the established role of fetal insulin secretion in fetal growth and macrosomia [117] and with previous reports on the associations between some T2D-risk alleles and reduced birth weight [118].

The findings outlined above suggest that the genetic and physiologic pathways that lead to GDM differ, at least in part, from those that lead to T2D. Future studies should assess whether these polygenic clusters based on physiological characteristics and pathways could be exploited in clinical care. As Powe et al. speculate [115], it is possible that, e.g., the pregnancy cluster 1 polygenic score (highest weighted glycemic traits including higher post-load glucose levels, lower disposition index, and higher adiposity measures) could identify women who are likely to benefit from early GDM screening. However, more studies in large and diverse cohorts with genetic and pregnancy phenotypic data are required to verify this. Of note, all currently available PRS/cluster studies have utilized glucose concentrations from OGTTs performed after 24 weeks’ gestation. An interesting future avenue would be to explore the genetic determinants of early pregnancy (<20 weeks’ gestation) glycemic traits in unselected and high-risk populations. In addition, studies combining maternal and offspring genetic information with novel metabolomic, proteomic, and other “omics” data obtained at different time points across gestation could bring new insights into the complex regulation of glycemic traits during pregnancy. 

## 6. Gene–Lifestyle Interactions with GDM and Postpartum Type 2 Diabetes

Although rapid environmental changes during recent decades mostly explain the current epidemic of both GDM and T2D, synergistic gene–environment interactions have also been speculated to play a role. In other words, the effects of our diabetogenic environment may at least partly depend on the genetic predisposition to the disease [119]. Randomized trials have shown that it is possible to delay or even prevent the development of T2D in individuals at elevated risk through lifestyle modification, focusing on weight loss, physical activity, and diet [120]. Studies of gene–environment interactions performed thus far suggest that the effectiveness of lifestyle modifications in reducing the incidence of T2D does not depend on the known underlying genetic predisposition [119].

### 6.1. Gene–Lifestyle Interactions with GDM

Whether the genetic risk in women with GDM modifies the effectiveness of lifestyle interventions is less well established. Barabash et al. [121] demonstrated a gene–lifestyle interaction between the *TCF7L2* rs7903146 polymorphism and the degree of adherence to a Mediterranean diet in relation to the onset of GDM. Only in T-risk allele carriers does the nutritional intervention modify the risk of developing GDM in a manner where women with moderate and high adherence had a reduced GDM risk compared to women with low adherence. Chen et al. [122] reported, in a follow-up setting, that women with prior GDM carrying a diabetes- and obesity-increasing *MC4R* genotype may obtain a greater improvement in insulin resistance through a lifestyle intervention than those with the non-risk genotype. Notably, the associations were independent of concurrent weight change. Furthermore, a *MTNR1B* polymorphism has been shown to modify the outcome of a lifestyle intervention aiming to prevent GDM in pregnant women [123]. Among Finnish women at high risk for GDM, only those not carrying the rs10830963 risk allele G seemed to benefit from a lifestyle intervention initiated in early pregnancy [123]. However, the sample size in this study was small, and this observation requires further replication. In addition, Van Poppel et al. [124] investigated whether interactions with *MTNR1B* polymorphisms are specific to healthy eating or physical activity interventions in the pan-European Vitamin D and Lifestyle Intervention (DALI) lifestyle trial. The lifestyle intervention did not affect the risk of developing GDM, and no interactions between *MTNR1B* rs10830962 and rs10830963 genotypes and the intervention were found. However, they demonstrated that in women homozygous for the risk allele of rs10830962, a physical activity intervention might be more beneficial than a healthy eating intervention for reducing maternal insulin resistance; cord blood C-peptide; and cord blood leptin, a proxy for neonatal adiposity. Thus, interactions between *MTNR1B* variants and physical activity deserve further investigation also due to the possible impact on neonatal outcomes. On the other hand, in the Tianjin Gestational Diabetes Mellitus Prevention Program, the *MTNR1B* variant rs10830962 did not show any interaction with a lifestyle intervention (healthy diet and physical activity) when assessed by changes in glycemic markers [125].

### 6.2. Interactions between PRS, Lifestyle Factors, and GDM/T2D

Only a few studies have explored possible interactions between PRSs and lifestyle. In the Diabetes Prevention Program study, the PRSs constructed with 34 SNPs previously associated with T2D were positively associated with a maternal history of GDM but did not modulate the response to a lifestyle intervention aimed at T2D prevention [113]. A recent study utilizing PRSs (50 SNPs associated with T2D) assessed the impact of the interaction between genetic risk and lifestyle intervention on the occurrence of GDM and postpartum T2D [126]. In their study, a high PRS was associated with GDM diagnosis and markers of abnormal glucose metabolism in mid- and late pregnancy and at 12 months postpartum. However, women within the highest T2D tertile of PRS were also those who most likely benefited from a lifestyle intervention aiming at the prevention of GDM during pregnancy and prevention of T2D during the first postpartum year. A very recent large prospective cohort study involving women with prior GDM, followed for 28 years, observed an inverse association between optimal lifestyle factors (non-smoking, healthy body mass index, healthy quality of diet, regular physical activity, and moderate alcohol consumption) and later-life T2D in a dose-dependent fashion [127]. Importantly, this association was independent of the underlying genetic risk characterized by 59 SNPs associated with T2D. Furthermore, Pagel et al. [128] investigated the association of PRSs (84 SNPs associated with T2D) and physical activity in existing GDM risk models. They found that increased physical activity was associated with decreased risk of GDM, and this risk reduction was particularly significant in individuals who were genetically predisposed to diabetes as assessed by PRS or family history.

## 7. Conclusions

In conclusion, research evidence from the past decade has demonstrated remarkable phenotypic and genotypic heterogeneity in women with GDM. Recent GWAS have confirmed the shared genetic architecture between GDM and T2D and shed light on the associated pathways leading to maternal dysglycemia, in addition to identifying a few novel genetic variants that appear more specific to the regulation of glucose metabolism in pregnancy. Together, genetic and clinical studies have also advanced our understanding of distinct GDM subtypes and their differing prognoses. However, significant research gaps remain concerning variations in paternal, fetal, and placental genetic factors that could contribute to the heterogeneity of GDM and its short- and long-term health implications in the mother and child. Considering the increasingly recognized impacts of the early pregnancy metabolic milieu on pregnancy/fetal outcomes, genetic determinants of first- and second-trimester glycemic and metabolic traits present an interesting area to be explored in future research. In addition, studies combining maternal and offspring genetic information with other “omics” data on maternal metabolism obtained across gestation could bring new insights into the complex regulation of glucose metabolism during pregnancy. With regards to gene–lifestyle interactions, there is some evidence that individuals with higher genetic risk may benefit more from adherence to a healthy diet or physical activity in GDM prevention. Findings such as this could facilitate the individualization of GDM treatment and the prevention of long-term metabolic sequelae. However, future studies should examine variants conferring individual differences in response to lifestyle interventions among women with GDM in more diverse populations and intervention trials with sufficient power and precisely measured lifestyle exposures.

## Figures and Tables

**Table 1 nutrients-14-04799-t001:** Genes associated with GDM in recent meta-analyses or GWAS.

Gene Symbol	Gene Functions/Associations *	Reference	Sample Size/Number of Included Studies	Type of Study
*CDKAL1*; CDK5 regulatory subunit associated protein 1 like 1	Encodes CDKAL1, a member of the methylthiotransferase family. The exact function of CDKALI has not been established but it has been implicated in glucose-stimulated insulin secretion [84,85] and risk variants associated with impaired insulin secretory capacity [51]. T2D-associated locus.	Kwak et al. [59]	Stage 1: GDM *n* = 468, Control *n* = 1242; Stage 2: GDM *n* = 931, Control *n* = 783	Two-stage GWAS
		Mao et al. [86]	GDM *n* = 10336; Control *n* = 17,445	Meta-analysis
		Zhang et al. [62]	29 studies	Meta-analysis
		Guo et al. [87]	14 studies	Meta-analysis
		Powe and Kwak [63]	23 studies, effective *n* = 2373–24,237	Meta-analysis
		Pervjakova et al. [61]	GDM *n* = 5485; control *n* = 347,856	Multi-ancestry GWAS
*CDKN2AB*; cyclin-dependent kinase inhibitor 2A/B	Two splice variants encode inhibitors of CDK. One alternate open reading frame product codes for a stabilizer of p53. Involved in regulation of cell proliferation and apoptosis. May protect pancreatic β-cells from glucotoxicity [88]. Risk variants associated with impaired insulin secretion from β-cells [89]. Mutations cause MODY2 [90]. MODY- and T2D-associated locus.	Zhang et al. [62]	29 studies	Meta-analysis
		Guo et al. [91]	14 studies	Meta-analysis
		Powe and Kwak [63]	23 studies; effective *n* = 2373–24,237	Meta-analysis
		Pervjakova et al. [61]	GDM *n* = 5485; control *n* = 347,856	GWAS
*GCK;* glucokinase	Encodes the GCK enzyme, a hexokinase with a role in glucose-stimulated insulin secretion in the pancreas, and glucose uptake and conversion to glycogen in the liver. T2D-associated locus.	Zhang et al. [62]	29 studies	Meta-analysis
		Mao et al. [86]	GDM *n* = 10336; control *n* = 17,445	Meta-analysis
*GLIS3;* GLIS family zinc finger 3	Encodes a nuclear protein with five C2H2-type zinc finger domains with a role in the development of the pancreas, thyroid, eye, liver, and kidney. Regulates insulin gene transcription, insulin secretion, and probably also β-cell survival [92]. T2D-associated locus.	Powe and Kwak [63]	23 studies, effective *n* = 2373–24,237	Meta-analysis
*HHEX/IDE;* hematopoietically expressed homeobox/insulin-degrading enzyme	HHEX codes a transcription factor belonging to the homebox family, implicated in developmental processes. IDE codes for insulin-degrading enzyme. Diabetes risk alleles linked to decreased pancreatic β-cell function [93]. T2D-associated locus.	Powe and Kwak [63]	23 studies, effective *n* = 2373–24,237	Meta-analysis
*HKDC1;* hexokinase-domain-containing 1	Codes for a zinc metallopeptidase that degrades intracellular insulin and other peptides (e.g., glucagon). Regulates glucose utilization and homeostasis, and may have a particular role during times of metabolic stress [94]. T2D-associated locus. Homeostasis during times of metabolic stress	Pervjakova et al. [61]	GDM *n* = 5485; control *n* = 347,856	GWAS
*HNF1A;* hepatocyte nuclear factor 1α	Encodes a transcription factor required for the expression of many liver-specific genes. Mutations in HNF1A causes MODY3 and common variants associated with T2D via affecting insulin secretion [95]. MODY- and T2D-associated locus.	Powe and Kwak [63]	23 studies, effective *n* = 2373–24,237	Meta-analysis
*IRS1;* insulin receptor substrate 1	Encodes a protein that is phosphorylated by insulin receptor tyrosine kinase. Associated with susceptibility to insulin resistance. T2D-associated locus.	Mao et al. [86]	GDM *n* = 10336; control *n* = 17,445	Meta-analysis
		Zhang et al. [62]	29 studies	Meta-analysis
*IGF2BP2;* insulin-like growth factor 2 mRNA-binding protein 2	Encodes a protein that regulates the translation of IGF2 mRNA. IGF2 is a polypeptide growth factor involved in the stimulation of insulin action [96]. T2D-associated locus.	Mao et al. [86]	GDM *n* = 10336; control *n* = 17,445	Meta-analysis
		Zhang et al. [62]	29 studies	Meta-analysis
		Wu et al. [97]	GDM *n* = 8204; control *n* = 15,221	Meta-analysis
*KCNJ11;* potassium inwardly rectifying channel subfamily J member 1	Codes for an integral membrane protein and inward-rectifier-type potassium channel involved in insulin release [98]. Mutations cause MODY13 [99]. MODY- and T2D-associated locus.	Mao et al. [86]	GDM *n* = 10336; control *n* = 17,445	Meta-analysis
		Zhang et al. [62]	29 studies	Meta-analysis
*KCNQ1;* potassium voltage-gated channel subfamily Q member 1	Encodes a voltage-gated potassium channel. Risk variants are likely to affect diabetes risk via changes in pancreatic expression of this protein and effects on insulin secretion [100,101]. T2D-associated locus.	Mao et al. [86]	GDM *n* = 10336; control *n* = 17,445	Meta-analysis
*MTNR1B;* melatonin receptor 1B	Encodes a high-affinity receptor for melatonin, which influences insulin secretion [102] in addition to other functions, such as regulating circadian rhythms. Variants associated with impaired insulin secretion [103]. T2D-associated locus.	Mao et al. [86]	GDM *n* = 10336; control *n* = 17,445	Meta-analysis
		Zhang et al. [62]	29 studies	Meta-analysis
		Wu et al. [97]	GDM *n* = 8204; control *n* = 15,221	Meta-analysis
		Powe and Kwak [63]	23 studies, effective *n* = 2373–24,237	Meta-analysis
		Pervjakova et al. [61]	GDM *n* = 5485; control *n* = 347,856	GWAS
*TCF7L2;* transcription factor 7-like 2	Codes for a high mobility group (box-containing transcription factor involved in the Wnt signaling pathway). Implicated in insulin production and processing [104]. T2D-associated locus.	Zhang et al. [62]	29 studies	Meta-analysis
		Mao et al. [86]	GDM *n* = 10336; control *n* = 17,445	Meta-analysis
		Wu et al. [97]	GDM *n* = 8204; control *n* = 15,221	Meta-analysis
		Pervjakova et al. [61]	GDM *n* = 5485; control *n* = 347,856	GWAS

* https://www.ncbi.nlm.nih.gov/clinvar/ (accessed on 5 November 2022) and https://www.proteinatlas.org/ (accessed on 5 November 2022). GDM = gestational diabetes; MODY = maturity-onset diabetes of the young; T2D = type 2 diabetes.

## Data Availability

Not applicable.

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
