# Peer review of "Genetic Risk Factors and Gene–Lifestyle Interactions in Gestational Diabetes"

_nutrients, 2022, doi:10.3390/nu14224799_

Round 1

Reviewer 1 Report

I don't find any major issues with this review article except a few spelling, period, and spacing errors. In table 1 few lines are not legible. one major concern with this review is how well the authors reviewed the diseases associated with Gestational diabetes. I recommend still a deep review is required. For example, retinopathy of prematurity is a juvenile eye disease associated with gestational diabetes is mentioned. kindly pay more attention 

Author Response

We appreciate the thorough review and constructive feedback on our manuscript. Please find our detailed point-by-point responses to both Reviewers attached. The changes made in the manuscript have been marked with the “track changes” function of MS Word. We are willing to modify the manuscript further if more changes are regarded as necessary.

Table 1 has been modified and all lines should be now legible. We kindly ask Reviewer #1 to elaborate the comment on comorbidities. In our manuscript, we do not focus on diseases associated with GDM or mention any of them. Thus, we would need further information how to modify our manuscript in this sense.

Reviewer 2 Report

Summary

With their review Jääskeläinen & Klemetti are providing a contemporary research update on whether thus-far identified genetic risk factors have use in predicting GDM risk, allow to differentiate GDM risk genotypes effectively, predict GDM sequelae and predict whether life-style interventions may change GDM risk. The authors discuss single genetic risk factors as well as polygenic risk scores.

In their review, the authors provide a comprehensive and well-balanced update of the role of genetics in GDM and the (possible) use of genetic information in GDM risk management.

General comments

The authors delivered a very matter-of-fact review.

There is scope to be clearer on how genetic risk factors are used / can be used to define different GDM genotypes.

There is scope to be clearer in stating which genetic risk factors are well established and which still need further corroboration. In this context, a more in-depth qualification of the PRS literature in terms of SNPs that are recurrent in the different PRS would be helpful in terms of appraising robustness of current PRS.

The paper would also benefit from clearly differentiating literature on predicting and/or modifying GDM risk from literature about predicting / modifying risk for future morbidity in women with prior GDM. The authors should consider the use of additional sub-headings.

The manuscript would benefit from some grammar and word-choice review by a native English speaker to make the language more precise throughout.

Comments on Abstract

Whereas the abstract clearly states that the authors will “point out” research gaps, this topic is not well developed in the manuscript. The authors should or remove this cue in the abstract or elaborate on the research gaps. The latter would allow for the authors to infuse some “opinion” into the manuscript.

Comments Section 1

- Line 52-53: add the comparative prevalence of T2D in women without GDM exposure over the same follow-up time to contrast the difference and highlight the strong disposition to T2D following GDM.

Comments Section 2:

- it would be helpful to clarify what is meant by “early GDM” vs, "late-onset" GDM in terms of gestational age.

Comments Section 3:

- A more clear differentiation of variants which are common with T2D and / or T1D and these that are specific / more strongly associating with GDM in a table would be beneficial to the reader.

- lines 148-164: this whole section is about PRS and should be moved to section 5. This will also link up better the final paragraph of this section (lines 166-170) with the rest of the section.

Comments Section 4:

None

Comments Section 5:

- Integrate Section 3 paragraph (lines 148-164) – elaborate a bit more on the clusters ID-ed in Powe et al as an example of using genetic risk factor to resolve genotypic heterogeneity; if possible add some more commentary on the pregnancy cluster.

- add subtitles to highlight difference in scope between GDM risk prediction and the prediction of T2D in women with a history of GDM. Apply this grouping in Table 2 as well; consider splitting in 2 tables. Include the PRS as elaborated in lines 148-164 in the GDM table.

- a more detailed review of the different PRS in terms of common SNPs as reported in the associated Table 2 would help the reader to appreciate the maturity of PRS as a tool to predict GDM / prognose sequelae.

Comments Section 6:

 - add subtitles to highlight difference in scope between literature discussing gene-lifestyle interactions in the context of modifying GDM risk and gene-life style interactions in the women with prior GDM and modification of the development of GDM sequelae like T2D.  prediction of T2D in women with a history of GDM.

Comments Section 7:

Add some paragraphs clearly highlighting what areas need further research and if possible add some putative directions of research.

Comments Table 1

The text in several cells is not readable as the row heights are not properly adjusted throughout. Ensure proper text alignment throughout the table

- Row 5: GLIS3 in Gene function column: correct: …development of pancreatic the thyroid… to development of the pancreas, the thyroid, …  

- All rows referring to Powe and Kwak; add the correct effective sample sizes per gene – SNP pair

Author Response

We appreciate the thorough review and constructive feedback on our manuscript. Please find our detailed point-by-point responses below and in the attached Rebuttal document. The changes made in the manuscript have been marked with the “track changes” function of MS Word. We are willing to modify the manuscript further if more changes are regarded as necessary.

Summary

With their review Jääskeläinen & Klemetti are providing a contemporary research update on whether thus-far identified genetic risk factors have use in predicting GDM risk, allow to differentiate GDM risk genotypes effectively, predict GDM sequelae and predict whether life-style interventions may change GDM risk. The authors discuss single genetic risk factors as well as polygenic risk scores.

In their review, the authors provide a comprehensive and well-balanced update of the role of genetics in GDM and the (possible) use of genetic information in GDM risk management.

General comments

The authors delivered a very matter-of-fact review.

There is scope to be clearer on how genetic risk factors are used / can be used to define different GDM genotypes.

There is scope to be clearer in stating which genetic risk factors are well established and which still need further corroboration. In this context, a more in-depth qualification of the PRS literature in terms of SNPs that are recurrent in the different PRS would be helpful in terms of appraising robustness of current PRS.

The paper would also benefit from clearly differentiating literature on predicting and/or modifying GDM risk from literature about predicting/modifying risk for future morbidity in women with prior GDM. The authors should consider the use of additional sub-headings.

The manuscript would benefit from some grammar and word-choice review by a native English speaker to make the language more precise throughout.

Response:  We thank the referee for these very relevant comments. We have now addressed these issues in our corrected manuscript (as specified in more detail in our responses below). We have also added subheadings and checked/corrected the grammar and punctuation.  

Comments on Abstract

Whereas the abstract clearly states that the authors will “point out” research gaps, this topic is not well developed in the manuscript. The authors should or remove this cue in the abstract or elaborate on the research gaps. The latter would allow for the authors to infuse some “opinion” into the manuscript.

Response: We agree with the Reviewer that this aspect was weak in the first version of our manuscript. We have now elaborated on the research gaps throughout the manuscript and in the conclusions paragraph. For instance, the following research gaps are pointed out:

Page 3:

“However, there is a gap in research evidence with regard to normal and abnormal glucose metabolism in early pregnancy (40, 41), with no evidence-based criteria for the diagnosis of early-onset GDM.”

Page 5:

 “However, even larger consortium studies should be carried out in the future to reach the statistical power needed to identify rarer genetic factors that may predispose to or protect against GDM.”

Page 6:

“Nevertheless, relatively few studies to date have investigated paternal, fetal, and placental genetic factors that affect maternal glucose metabolism and/or GDM risk.” “However, the mechanisms of molecular and genetic crosstalk within the mother-placental-fetus triad, implicated in glucose homeostasis during pregnancy, are a largely unknown field that should be addressed in future research.”   

Page 12:

The above-outlined findings suggest that the genetic and physiologic pathways that lead to GDM differ, at least in part, from those that lead to T2D. Future studies should assess whether these polygenic clusters based on physiological characteristics and pathways could be exploited in clinical care.”

An interesting future avenue would be to explore the genetic determinants of early-pregnancy (<20 weeks’ gestation) glycemic traits in both unselected and high-risk populations.”

In addition, studies combining maternal and offspring genetic information with novel metabolomic, proteomic and other “omics”  data obtained at different time points across gestation could bring new insights into the complex regulation of glycemic traits during pregnancy”

Page 13-14:

“Thus, interactions between MTNR1B variants and physical activity deserve      further investigation also due to the possible impact on neonatal outcomes.”

Comments Section 1

- Line 52-53: add the comparative prevalence of T2D in women without GDM exposure over the same follow-up time to contrast the difference and highlight the strong disposition to T2D following GDM.

Response: This addition has been made as requested. The sentence now reads: “In a recent Finnish study with a 23-year follow-up, the prevalence of T2D increased linearly over time, reaching 50.4% in women with prior GDM (vs. 5.5% in women without GDM) by the end of the study period (14). Type 1 diabetes (T1D), on the other hand, developed in 5.7% of Finnish women within 7 years after a GDM pregnancy, whereas no T1D cases were identified in the control group without GDM (14)”.

Comments Section 2:

- it would be helpful to clarify what is meant by “early GDM” vs, "late-onset" GDM in terms of gestational age.

Response: We have now clarified that by “early GDM” we mean GDM diagnosed before 20 weeks’ gestation and with “late GDM” we mean GDM diagnosed between 24-30 weeks’ gestation.

Comments Section 3:

- A more clear differentiation of variants which are common with T2D and / or T1D and these that are specific / more strongly associated with GDM in a table would be beneficial to the reader.

Response: Table 1 was already very large and wide. Therefore, after careful consideration, we decided that we will not add a new column. However, we have now mentioned which type of diabetes (in addition to GDM) has been associated with each of the variants in the “gene functions” column of Table 1, which has now been renamed “gene functions/associations”. In the manuscript text, on page 4 (at the end of second paragraph in the section subtitled “Maternal genetic risk factors of GDM”) we have also added some sentences discussing these issues.

- lines 148-164: this whole section is about PRS and should be moved to section 5. This will also link up better the final paragraph of this section (lines 166-170) with the rest of the section.

Response: We agree with the Reviewer and have now made this change.

Comments Section 5:

- Integrate Section 3 paragraph (lines 148-164) – elaborate a bit more on the clusters IDed in Powe et al as an example of using genetic risk factor to resolve genotypic heterogeneity; if possible add some more commentary on the pregnancy cluster.

Response: We have now integrated the section from the Section 3 to the Section 5. Furthermore, we have added some discussion on the potential clinical significance of the results by Powe et al.

- add subtitles to highlight difference in scope between GDM risk prediction and the prediction of T2D in women with a history of GDM. Apply this grouping in Table 2 as well; consider splitting in 2 tables. Include the PRS as elaborated in lines 148-164 in the GDM table.

Response: We have now added subtitles “PRSs and GDM” and “PRS and T2D” within the text. We have also restructured and modified Table 2 by adding separate column titles for different outcomes. 

- a more detailed review of the different PRS in terms of common SNPs as reported in the associated Table 2 would help the reader to appreciate the maturity of PRS as a tool to predict GDM / prognose sequelae.

Response: We have now added this sentence: “Most of the SNPs used in these PRS studies have typically been significantly associated (P < 5x10−8) with T2D in many GWASs in the general population.”

Comments Section 6:

 - add subtitles to highlight difference in scope between literature discussing gene-lifestyle interactions in the context of modifying GDM risk and gene-life style interactions in the women with prior GDM and modification of the development of GDM sequelae like T2D.  prediction of T2D in women with a history of GDM.

Response: We have now added the following subtitles: “Gene-lifestyle interactions with GDM” and “Interactions between PRS and lifestyle factors with GDM and T2D”. We hope that these changes have clarified the section.

Comments Section 7:

Add some paragraphs clearly highlighting what areas need further research and if possible add some putative directions of research.

Response: As mentioned already above, we have added a few more sentences pointing out areas that need further research throughout the manuscript and summarized them in the Conclusions paragraph.

Comments Table 1

The text in several cells is not readable as the row heights are not properly adjusted throughout. Ensure proper text alignment throughout the table.

Response: In our submitted Word version the lines are all visible. However, we have now tried to make sure that the lines are also visible in the pdf produced from the Word document.

- Row 5: GLIS3 in Gene function column: correct: …development of pancreatic the thyroid… to development of the pancreas, the thyroid, … 

- All rows referring to Powe and Kwak; add the correct effective sample sizes per gene – SNP pair

Response: Thank you for these corrections, we have now implemented the first one. However, referring to Powe and Kwak in Table 1, it is very difficult to fit all the effective sample sizes per gene-SNP pair. Hence, we have not implemented this suggestion. Unfortunately, we could not identify specific total sample sizes for “control” and “GDM” groups in this meta-analysis in the same way as we did for the other studies
